# Plan-Seq-Learn: Language Model Guided RL for Solving Long Horizon Robotics Tasks

**Abstract:** Large Language Models (LLMs) are highly capable of performing planning for long-horizon robotics tasks, yet existing methods require access to a pre-defined skill library (*e.g.* picking, placing, pulling, pushing, navigating). However, LLM planning does not address how to design or learn those behaviors, which remains challenging particularly in long-horizon settings. Furthermore, for many tasks of interest, the robot needs to be able to adjust its behavior in a fine-grained manner, requiring the agent to be capable of modifying *low-level* control actions. Can we instead use the internet-scale knowledge from LLMs for high-level policies, guiding reinforcement learning (RL) policies to efficiently solve robotic control tasks online without requiring a pre-determined set of skills? In this paper, we propose **Plan-Seq-Learn** (PSL): a modular approach that uses motion planning to bridge the gap between abstract language and learned low-level control for solving long-horizon robotics tasks from scratch. We demonstrate that PSL is capable of solving 20+ challenging single and multi-stage robotics tasks on four benchmarks at success rates of over 80% from raw visual input, out-performing language-based, classical, and end-to-end approaches. Video results and code at https://planseqlearn.github.io/.

## 1 Introduction

In recent years, the field of robot learning has witnessed a significant transformation with the emergence of Large Language Models (LLMs) as a mechanism for injecting internet-scale knowledge into robotics. One paradigm that has been particularly effective is LLM planning over a predefined set of skills [1, 2, 3, 4], producing strong results across a wide range of robotics tasks. These works assume the availability of a pre-defined skill library that abstracts away the robotic control problem. They instead focus on designing methods to select the right sequence skills to solve a given task. However, for robotics tasks involving contact-rich robotic manipulation (Fig. 1), such skills are often not available, require significant engineering effort to design or train a-priori or are simply not expressive enough to address the task. How can we move beyond pre-built skill libraries and enable the application of language models to general purpose robotics tasks with as few assumptions as possible? Robotic systems need to be capable of **online improvement** over *low-level* control policies while being able to **plan** over long horizons.

End-to-end reinforcement learning (RL) is one paradigm that can produce complex low-level control strategies on robots with minimal assumptions [5, 6, 7, 8, 9, 10, 11]. However, RL methods are traditionally limited to the short horizon regime due to the significant challenge of exploration in RL, especially in high-dimensional continuous action spaces characteristic of robotics tasks. RL methods struggle with longer-horizon tasks in which high-level reasoning and low-level control must be learned simultaneously; effectively decomposing tasks into sub-sequences and accurately achieving them is challenging in general [12, 13].

Our key insight is that LLMs and RL have *complementary* strengths and weaknesses. Language models can leverage internet scale knowledge to break down long-horizon tasks [1, 14] into achievable sub-goals, but lack a mechanism to produce low-level robot control strategies [15], while RL can

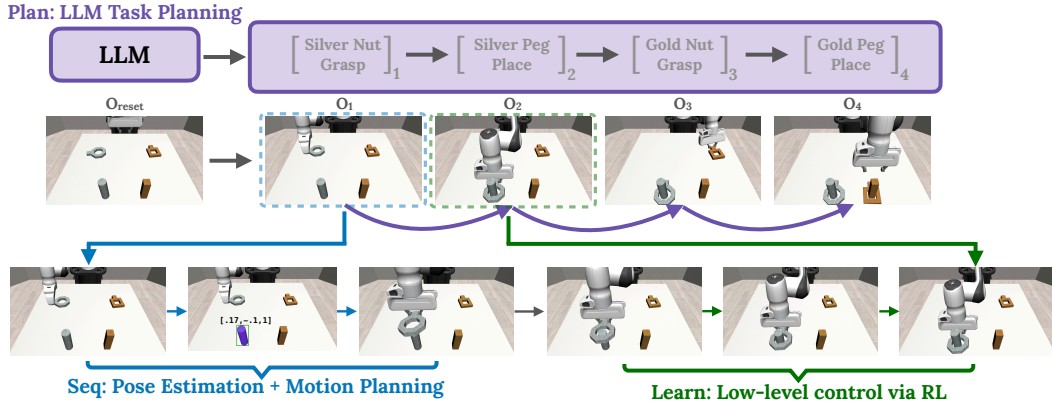

Figure 1: **Long horizon task visualization.** We visualize PSL solving the NutAssembly task, in which the goal is to put both nuts on their respective pegs. After predicting the high-level plan using an LLM, PSL computes a target robot pose, achieves it using motion planning and then learns interaction via RL (*third row*).

discover complex control behaviors on robots but struggles to simultaneously perform long-term reasoning [16]. However, directly combining the two paradigms, for example, via training a language conditioned policy to solve a new task, does not address the exploration problem. The RL agent must now simultaneously learn language semantics and low-level control. Ideally, the RL agent should be able to follow the guidance of the LLM, enabling it to learn to efficiently solve each predicted sub-task online. How can we connect the abstract language space of an LLM with the low-level control space of the RL agent in order to address the long-horizon robot control problem?

In this work, we propose a learning method to solve long-horizon robotics tasks by tracking language model plans using motion planning and learned low-level control. Our approach, called **P**lan-**S**eq-**L**earn (PSL), is a modular framework in which a high-level language plan given by an LLM (**Plan**) is interpreted and executed using motion planning (**Seq**), enabling the RL policy (**Learn**) to rapidly learn short-horizon control strategies to solve the overall task. This decomposition enables us to effectively leverage the complementary strengths of each module: language models for abstract planning, vision-based motion planning for task plan tracking as well as achieving robot states and RL policies for learning low-level control. Furthermore, we improve learning speed and training stability by sharing the learned RL policy across all stages of the task, using local observations for efficient generalization, and introducing a simple, yet scalable curriculum learning strategy for tracking the language model plan. To our knowledge, ours is the first work enabling language guided RL agents to efficiently learn low-level control strategies for long-horizon robotics tasks.

Our contributions are: 1) A novel method for long-horizon robot learning that tightly integrates large language models for high-level planning, motion planning for skill sequencing and RL for learning low-level robot control strategies; 2) Strategies for efficient policy learning from high-level plans, which include policy observation space design for locality, shared policy network and reward function structures, and curricula for stage-wise policy training; 3) An extensive experimental evaluation demonstrating that PSL can solve **20+** long-horizon robotics tasks, outperforming SOTA baselines across four benchmark suites at success rates of **over 80%** purely from visual input. PSL produces agents that solve challenging long-horizon tasks such as NutAssembly at **over 95%** success rate.

## 2 Plan-Seq-Learn

In this section, we describe our method for solving long-horizon robotics tasks, PSL, outlined in Fig. 2. Given a text description of the task, our method breaks up the task into meaningful sub-sequences (**Plan**), uses vision and motion planning to translate sub-sequences into initialization regions (**Seq**) from which we can efficiently train local control policies using RL (**Learn**).

### 2.1 Related Work

LLMs have been applied to RL and robotics in a wide variety of ways, from planning [1, 2, 14, 3, 4, 17, 18, 19], reward definition [20, 21], generating quadrupedal contact-points [22], producing tasks

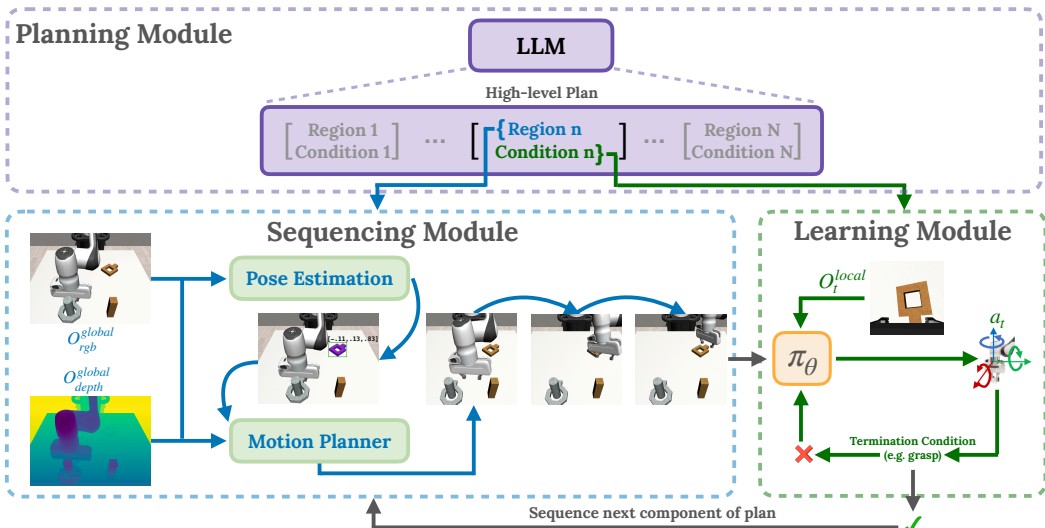

Figure 2: **Method overview.** PSL decomposes tasks into a list of regions and stage termination conditions using an LLM (*top*), sequences the plan using motion planning (*left*) and learns control policies using RL (*right*).

for policy learning [23, 24] and controlling simulation-based trajectory generators to produce diverse tasks [25]. Our work instead focuses on the online learning setting and aims to leverage language model driven planning to guide RL agents to solve new robotics tasks in a sample efficient manner. BOSS Zhang et al. [26] is closest to our overall method; this concurrent work also leverages LLM guidance to learn new skills via RL. Crucially, their method depends on the existence of a skill library and learns skills that are combination of high-level actions. Our method instead efficiently learns *low-level* robot control skills without depending on a pre-defined skill library, by taking advantage of motion planning to track an LLM plan. We include a more detailed description of the related work including connections to classical planning literature as well as integrated planning and learning methods in Appendix H.

## 2.2 Problem Setup

We consider Partially Observed Markov Decision Processes (POMDP) of the form $(\mathcal{S}, \mathcal{A}, \mathcal{T}, \mathcal{R}, p_0, \mathcal{O}, p_O, \gamma)$. $\mathcal{S}$ is the set of environment states, $\mathcal{A}$ is the set of actions, $\mathcal{T}(s' \mid s, a)$ is the transition probability distribution, $\mathcal{R}(s, a, s')$ is the reward function, $p_0$ is the distribution over the initial state $s_0 \sim p_0$, $\mathcal{O}$ is the set of observations, $p_O$ is the distribution over observations conditioned on the state $O \sim p_O(O|s)$ and $\gamma$ is the discount factor. In our case, the observation space is the set of all RGB-D (RGB and depth) images. The reward function is defined by the environment. The agent's goal is to maximize the expected sum of rewards over the trajectory, $\mathbb{E}\left[\sum_t \gamma^t \mathcal{R}(s_t, a_t, s_{t+1})\right]$. In our work, we consider POMDPs that describe an embodied robot agent interacting with a scene. We assume that a text description of the task, $g_l$, is provided to the agent in natural language.

## 2.3 Overview

To solve long-horizon robotics tasks, we need a module capable of bridging the gap between zero-shot language model planning and learned low-level control. Observe that many tasks of interest can be decomposed into alternating phases of contact-free motion and contact-rich interaction. One first approaches a target region and then performs interaction behavior, prior to moving to the next sub-task. Contact-free motion generation is exactly the motion planning problem. For estimating the position of the target region, we note that state-of-the-art vision models are capable of accurate language-conditioned state estimation [27, 28, 29, 30, 31, 32]. As a result, we propose a Sequencing Module which uses off-the-shelf vision models to estimate target robot states from the language plan and then achieves these states using a motion planner. From such states, we train interaction policies that optimize the task reward using RL. See Alg. 1 and Fig. 2 for an overview of our method.

## 2.4 Planning Module: Zero-Shot High-level Planning

Long-horizon tasks can be broken into a series of stages to execute. Rather than discovering these stages using interaction or using a task planner [33] that may require privileged information about the

environment, we use language models to produce natural language plans zero shot without access to the environment. Specifically, given a task description $g_l$ by a human, we prompt an LLM to produce a plan. Designing the plan granularity and scope are crucial; we need plans that can be interpreted by the Sequencing Module, a vision-based system that produces and achieves robot poses using motion planning. As a result, the LLM predicts a target region (a natural language label of an object/receptacle in the scene, e.g. "silver peg") which can be translated into a target pose to achieve at the beginning of each stage of the plan.

When the RL policy is executing a step of the plan, we propose to add a stage termination condition (e.g. *grasped*, *placed*, etc.) to know the stage is complete and to move onto the next stage. These stage termination conditions are estimated using vision. We describe the stage termination conditions in greater detail in Sec. 2.6 and Appendix D. The LLM prompt consists of the task description $g_l$, the list of supported stage termination conditions (which we hold constant across all environments) and additional prompting strings for output formatting. We format the language plans as follows: ("Region 1", "Termination Condition 1"), ... ("Region N", "Termination Condition N"), assuming the LLM predicts N stages. Below, we include an example prompt and plan for the Nut Assembly task.

---

**Prompt:** Stage termination conditions: (grasp, place). Task description: The silver nut goes on the silver peg and the gold nut goes on the gold peg. Give me a simple plan to solve the task using only the stage termination conditions. Make sure the plan follows the formatting specified below and make sure to take into account object geometry. Formatting of output: a list in which each element looks like: (<object/region>, <operator>). Don't output anything else.
**Plan:** [("silver nut","grasp"), ("silver peg", "place"), ("gold nut", "grasp"), ("gold peg", "place")]

---

While any language model can be used to perform this planning process, we found that of a variety of publicly available LLMs (via weights or API), only GPT-4 [34] was capable of producing correct plans across all the tasks we consider. We provide additional details in Appendix D and example prompts in Appendix G.

### 2.5 Sequencing Module: Vision-based Plan Tracking

Given a high-level language plan, we now wish to step through the plan and enable a learned RL policy to solve the task, using off-the-shelf vision to produce target poses for a motion planning system to achieve. At stage X of the high-level plan, the Sequencing Module takes in the corresponding step high-level plan ("Region Y", "Termination Condition Z") as well as the current global observation of the scene $O^{global}$ (RGB-D view(s) that cover the whole scene), predicts a target robot pose $q_{target}$ and then reaches the robot pose using motion planning.

**Vision and Estimation:** Using a text label of the target region of interest from the high-level plan and observation $O^{global}$, we need to compute a target robot state $q_{target}$ for the motion planner to achieve. In principle, we can train an RL policy to solve this task (learn a policy $\pi_v$ to map $O^{global}$ to $q_{target}$) given the environment reward function. However, observe that the 3D position of the target region is a reasonable estimate of the optimal policy $\pi_v^*$ for this task: intuitively, we wish to initialize the robot nearby to the region of interest so it can efficiently learn interaction. Thus, we can bypass learning a policy for this step by leveraging a vision model to estimate the 3D coordinates of the target region. We opt to use Segment Anything [27] to perform segmentation, as it is capable of recognizing a wide array of objects, and use calibrated depth images to estimate the coordinates of the target region. We convert the estimated region pose into a target robot pose $q_{target}$ for motion planning using inverse kinematics.

**Motion Planning:** Given a robot start configuration $q_0$ and a robot goal configuration $q_{target}$ of a robot, the motion planning module aims to find a trajectory of way-points $\tau$ that form a collision-free path between $q_0$ and $q_{target}$. For manipulation tasks, for example, $q$ represents the joint angles of a robot arm. We can use motion planning to solve this problem directly, such as search-based planning [35], sampling-based planning [36] or trajectory optimization [37]. In our

implementation, we use AIT* [38], a sampling-based planner, due to its minimal setup requirements (only collision-checking) and favorable performance on planning. For implementation details, please see Appendix D.

Overall, the Sequencing Module functions as the connective tissue between language and control by moving the robot to regions of interest in the plan, enabling the RL agent to quickly learn short-horizon interaction behaviors to solve the task.

## 2.6 Learning Module: Efficiently Learning Local Control

Once the agent steps through the plan and achieves states near target regions of interest, it needs to train an RL policy $\pi_\theta$ to learn low-level control for solving the task. We train $\pi_\theta$ using DRQ-v2 [39], a SOTA visual model-free RL algorithm, to produce low-level control actions (joint control or end-effector control) from images. Furthermore, we propose three modifications to the learning pipeline in order to further improve learning speed and stability.

First, we train a *single* RL policy across all stages, stepping through the language plan via the Sequencing Module, to optimize the task reward function. The alternative, training a separate policy per stage, would require designing stage specific reward functions per task. Instead, our design enables the agent to solve the task using a single reward function by sharing the policy and value functions across stages. This simplifies the training setup and allowing the agent to account for future decisions as well as inaccuracies in the Sequencing Module. For example, if $\pi_\theta$ is initialized at a sub-optimal position relative to the target region, $\pi_\theta$ can adapt its behavior according to its value function, which is trained to model the full task return $\mathbb{E}\left[\sum_t \gamma^t \mathcal{R}(s_t, a_t, s_{t+1})\right]$.

Second, instead of executing $\pi_\theta$ for a fixed number of steps per stage $H_l$, we predict a stage termination condition using the language model and evaluate the condition at every time-step to test if a stage is complete, otherwise it times out after $H_l$ steps. This process functions as a form of curriculum learning: only once a stage is completed is the agent allowed to progress to the next stage of the plan. As we ablate in Sec. 4, stage termination conditions enable the agent to learn more performant policies by preventing dithering behavior at each stage. For the tasks we consider, stage termination conditions involve checking for grasping or placement. As an example, in the nut assembly task shown in Fig. 1, once $\pi_\theta$ places the silver nut on the silver peg, the placement condition triggers and the Sequencing Module moves the arm to near the gold peg.

Finally, as opposed to training the policy using the global view of the scene ($O^{global}$), we train using *local* observations $O^{local}$, which can only observe the scene in a small region around the robot (*e.g.* wrist camera views for robotic manipulation). This design choice affords several unique properties that we validate in Appendix C, namely: 1) improved learning efficiency and speed, 2) ease of chaining pre-trained policies. Our policies are capable of leveraging local views because of the decomposition in PSL: the RL policy simply has to learn interaction behaviors in a small region, it has no need for a global view of the scene, in contrast to an end-to-end RL agent that would need to see a global view of the scene to know where to go to solve a task. For additional details in regarding the structure and training process of the Learning Module, see Appendix D.

# 3 Experimental Setup

## 3.1 Tasks

We conduct experiments on single and multi-stage robotics tasks across four simulated environment suites (**Meta-World**, **Obstructed Suite**, **Kitchen** and **Robosuite**) which contain obstructed settings, contact-rich setups, and sparse rewards (Fig. F.1). See Appendix F for additional details.

**Meta-World:** [40] is an RL benchmark with a rich source of tasks. From Meta-World, we select four long-horizon tasks: MW-Disassemble (removing a nut from a peg), MW-BinPick (picking and placing a cube), MW-Assembly (picking and placing a nut on peg), MW-Hammer (grasp a hammer and hitting a nail).

**ObstructedSuite:** Yamada et al. [41] contains tasks that evaluate our agent's ability to plan, move and interact with the environment in the presence of obstacles. It consists of three tasks:

`OS-Lift` (lift a cube in a tall box), `OS-Push` (push a block surrounded by walls), and `OS-Assembly` (avoiding obstacles to place table leg at target).

**Kitchen:** [42, 43] tests two aspects of our agent: its ability to handle sparse terminal rewards and its long-horizon manipulation capabilities. The single-stage kitchen tasks include `K-Slide` (push slide cabinet to the right), `K-Kettle` (place kettle on back stove), `K-Burner` (turn burner knob), `K-Light` (flick light switch to "on"), and `K-Microwave` (open microwave door). The multi-stage Kitchen tasks denote the number of stages in the name and include combinations of the aforementioned single tasks.

**Robosuite:** [44] contains a wide array of robotic manipulation tasks ranging from single stage (`RS-Lift` - lift a cube, `RS-Door` - open a door) to multi-stage (`RS-NutRound`,`RS-NutSquare`, `RS-NutAssembly` - pick-place nut(s) onto target peg(s) and `RS-Bread`, `RS-Cereal`, `RS-Milk`, `RS-Can`, `RS-CerealMilk`, `RS-CanBread` - pick-place object(s) into appropriate bin(s)). Unlike the other environment suites, which simplify aspects of the low-level control, Robosuite emphasizes realism and fidelity to real-world control, enabling us to highlight the potential of our method to be applied to real systems.

## 3.2 Baselines

We compare against two types of baselines, methods that learn from data and methods that perform offline planning. We include additional details in Appendix D.

**Learning Methods. E2E:** [39] DRQ-v2 is a SOTA model-free visual RL algorithm also used to train our low-level control policy. **RAPS:** [45] is a hierarchical RL method that modifies the action space of the agent with engineered subroutines (primitives). RAPS greatly accelerates learning speed, but is limited in expressivity due to its action space, unlike PSL. **MoPA-RL:** [41] is similar to PSL in its integration of motion planning and RL but differs in that it does not leverage a task planner; it uses the RL agent to decide when and where to call the motion planner. In initial experiments, we found that MoPA-RL failed to learn with visual input; we instead use reported numbers from the paper from experiments using privileged state information on the Obstructed Suite of tasks.

**Planning Methods.TAMP:** [46] is a classical baseline that uses a privileged view of the world to perform joint high-level (task planning) and low-level planning (motion planning with primitives) for solving long-horizon robotics tasks. **SayCan:** a re-implementation of SayCan [1] using publicly available LLMs that performs LLM planning with a fixed set of pre-defined skills. Following the SayCan paper, we specify a skill library consisting of object picking and placing behaviors using pose-estimation, motion-planning and heuristic action primitives. We do not learn the pick skill as done in SayCan because our setup does not contain a separate set of train and evaluation environments. In this work, we evaluate the single-task RL regime in which the agent is tested with held out poses, not held out environments.

## 3.3 Experiment details

We evaluate all methods aside from TAMP and MoPA-RL (which use privileged simulator information) using visual input. SayCan and PSL use $O^{global}$ and $O^{local}$. For E2E and RAPS, we provide the learner access to a single global fixed view observation from $O^{global}$ for simplicity and speed of execution, as we did not find meaningful performance improvement in these baselines by incorporating additional camera views. We measure performance in terms of task success rate with respect to the number of trials (episodes). We do so to provide a fair metric for evaluating a variety of different low-level control implementations across PSL, RAPS, and E2E. Each method is trained for 10K episodes total. We train on each task using the default reward function without modification. For each method, we run 7 seeds on every task and average across 10 evaluations.

## 4 Results

We begin by evaluating PSL on a variety of single stage tasks across Robosuite, Meta-World, Kitchen and ObstructedSuite. Next, we scale our evaluation to the long-horizon regime in which we show that

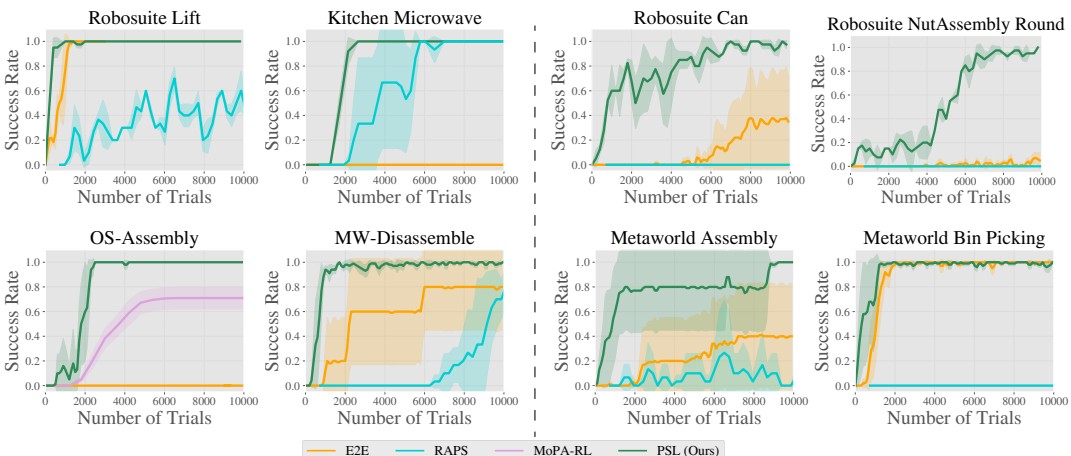

Figure 3: **Sample Efficiency Results.** We plot task success rate as a function of the number of trials. PSL improves on the sample efficiency of the baselines across each task in Robosuite, Kitchen, Meta-World, and Obstructed Suite. PSL is able to do so because it initializes the RL policy near the region of interest (as predicted by the Plan and Sequence Modules) and leverages local observations to efficiently learn interaction. Additional learning curves in Appendix C.

|  | RS-Bread | RS-Can | RS-Milk | RS-Cereal | RS-NutRound | RS-NutSquare |
|---|---|---|---|---|---|---|
| **E2E** | .52 ± .49 | 0.32 ± .44 | .02 ± .04 | 0.0 ± 0.0 | .06 ± .13 | 0.02 ± .045 |
| **RAPS** | 0.0 ± 0.0 | 0.0 ± 0.0 | 0.0 ± 0.0 | 0.0 ± 0.0 | 0.0 ± 0.0 | 0.0 ± 0.0 |
| **TAMP** | 0.9 ± .01 | 1.0 ± 0.0 | .85 ± .06 | **1.0 ± 0.0** | 0.4 ± 0.3 | .35 ± .2 |
| **SayCan** | .93 ± .09 | 1.0 ± 0.0 | 0.9 ± .05 | .63 ± .09 | .56 ± .25 | .27 ± .21 |
| **PSL** | **1.0 ± 0.0** | 1.0 ± 0.0 | **1.0 ± 0.0** | **1.0 ± 0.0** | **.98 ± .04** | **.97 ± .02** |

Table 1: **Robosuite Two Stage Results.** Performance is measured in terms of success rate on two-stage (*2 planner actions*) tasks. SayCan is competitive with PSL on pick-place style tasks, but SayCan's performance drops considerably (86.5% to 41.5% on average) on contact-rich tasks involving assembling nuts due to cascading failures. Online learning methods (E2E and RAPS) make little progress on the long-horizon tasks in Robosuite. On the other hand, PSL is able to solve each task with at least 97% success rate.

PSL can leverage LLM task planning to efficiently solve multi-stage tasks. We include additional experiments, ablations and analyses in Appendix C.

**PSL accelerates learning efficiency on a wide array of single-stage benchmark tasks.** For single-stage manipulation, (in which the LLM predicts only a single step in the plan), the Sequencing Module motion plans to the specified region, then hands off control to the RL agent to complete the task. In this setting, we solely evaluate the learning methods since the planning problem is trivial (only one step). We observe improvements in learning efficiency (with respect to number of trials) as well as final performance in comparison to the learning baselines E2E, RAPS and MoPA-RL, across 11 tasks in Robosuite, Meta-World, Kitchen and ObstructedSuite (Fig. 3, left). For all learning curves, please see the Appendix C. PSL especially performs well on sparse reward tasks, such as in Kitchen, for which a key challenge is figuring out which object to manipulate and where it is. Additionally, we observe qualitatively meaningful behavior using PSL: PSL learns to use the gripper to grasp and turn the burner knob, unlike E2E or RAPS which end up using other joints to flick the burner to the right position.

**PSL efficiently solves tasks with obstructions by leveraging motion planning.** We now consider three tasks from the Obstructed Suite in order to highlight PSL's effectiveness at learning control in the presence of obstacles. As we observe in Fig. 3 and Fig. C.2, PSL is able to do so efficiently, solving each task within 5K episodes, while E2E fails to make progress. PSL is able to do so because the Sequencing Module handles the obstacle avoidance implicitly via motion planning and initializes the RL policy in advantageous regions near the target object. In contrast, E2E spends a significant amount of time attempting to reach the object in spite of the obstacles, failing to learn the task. While MoPA-RL is also able to solve many of the tasks, it requires more trials than PSL even though it operates over *privileged* state input, as the agent must simultaneously learn *when* and *where* to motion plan as well as *how* to manipulate the object.

| Stages | RS-CerealMilk 4 | RS-CanBread 4 | RS-NutAssembly 4 | K-MS-3 3 | K-MS-4 4 | K-MS-5 5 |
|---|---|---|---|---|---|---|
| **E2E** | $0.0 \pm 0.0$ | $0.0 \pm 0.0$ | $0.0 \pm 0.0$ | $0.0 \pm 0.0$ | $0.0 \pm 0.0$ | $0.0 \pm 0.0$ |
| **RAPS** | $0.0 \pm 0.0$ | $0.0 \pm 0.0$ | $0.0 \pm 0.0$ | $.89 \pm 0.1$ | $0.3 \pm .15$ | $0.0 \pm 0.0$ |
| **TAMP** | $.71 \pm .05$ | $.72 \pm .25$ | $0.2 \pm 0.3$ | $1.0 \pm 0.0$ | $0.0 \pm 0.0$ | $0.0 \pm 0.0$ |
| **SayCan** | $.73 \pm .05$ | $.63 \pm .21$ | $.23 \pm .21$ | $1.0 \pm 0.0$ | $0.0 \pm 0.0$ | $0.0 \pm 0.0$ |
| **PSL** | $\mathbf{.85 \pm .21}$ | $\mathbf{0.9 \pm 0.2}$ | $\mathbf{.96 \pm .08}$ | $1.0 \pm 0.0$ | $\mathbf{.67 \pm .22}$ | $\mathbf{.67 \pm .22}$ |

Table 2: **Multistage (Long-horizon) results.** Performance is measured in terms of mean task success rate at convergence. PSL is the consistently solves each task, outperforming planning methods by over 70% on challenging contact-intensive tasks such as NutAssembly.

**PSL enables visuomotor policies to learn long-horizon behaviors with up to 5 stages.** Two-stage results across Robosuite and Meta-World are shown in Table 1 and Table C.3, with learning curves in Fig. 3 (right) and Fig. C.3. On the Robosuite tasks, E2E and RAPS fail to make progress: while they learn to reach the object, they fail to consistently grasp it, let alone learn to place it in the target location. On the Meta-World tasks, the learning baselines perform well on most tasks, achieving similar performance to PSL due to shaped rewards, simplified low-level control (no orientation changes) and small pose variations. However, PSL is significantly more sample-efficient than E2E and RAPS as shown in Fig. C.3. TAMP and SayCan are able to achieve high performance across each PickPlace variant of the Robosuite tasks ($93.75\%, 86.5\%$ averaged across tasks), as the manipulation skills do not require significant contact-rich interaction, reducing failure skill failure rates. Cascading failures still occur due to the baselines' open-loop nature of execution, imperfect state estimation (SayCan), planner stochasticity (TAMP). Only PSL is able to achieve perfect performance across each task, avoiding cascading failures by learning from online interaction.

On multi-stage tasks (involving 3-5 stages), we find that TAMP and SayCan performance drops significantly in comparison to PSL ($61\%, 51\%$ vs. $90\%$ averaged across tasks). For multiple stages, the cascading failure problem becomes all the more problematic, causing all three baselines to fail at intermediate stages, while PSL is able to learn to adapt to imperfect Sequencing Module behavior via RL. See Table 2 for a detailed breakdown of the results.

**PSL solves contact-rich, long-horizon control tasks such as NutAssembly.** In these experiments, we show that PSL can learn to solve contact-rich tasks (RS-NutRound, RS-NutSquare, RS-NutAssembly) that pose significant challenges for classical methods and LLMs with pre-trained skills due to the difficulty of designing manipulation behaviors under continuous contact. By learning an interaction policy whose purpose is to produce locally correct contact-rich behavior, we find that PSL is effective at performing contact-rich manipulation over long horizons (Table 1, Table 2), outperforming SayCan by a wide margin ($97\%$ vs. $35\%$ averaged across tasks). Our decomposition into contact-free motion generation and contact-rich interaction decouples the *what* (target nut) and *where* (peg) from the *how* (precision grasp and contact-rich place), allowing the RL agent to simply focus on the aspect of the problem that is challenging to estimate a-priori: how to interact with the objects in the appropriate manner.

## 5 Conclusions

In this work, we propose PSL, a method that integrates the long-horizon reasoning capabilities of language models with the dexterity of learned RL policies via a skill sequencing module. At the heart of our method lies the decomposition of robotics tasks into sequential phases of contact-free motion generation (using language model planning) and environment interaction. We solve these phases using motion planning (informed by visual pose-estimation) and model-free RL respectively, an approach which we validate via an extensive experimental evaluation. We outperform state-of-the-art methods for end-to-end RL, hierarchical RL, classical planning and LLM planning on over 20 challenging vision-based control tasks across four benchmark environment suites. In the future, this work could be extended to improving a pre-existing robot skill library over time using RL, enabling an agent to perform planning with an ever increasing repertoire of skills that can be refined at a low-level. PSL can also be applied to sim2real transfer, since the policies we train in this work use local observations, they are more amenable to sim2real transfer [11].

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

# Appendix

## A  Table of Contents

- **Ethics, Impacts and Limitations Statement** (Appendix B): Statement addressing potential ethics concerns and impacts as well as limitations of our method.
- **Additional Experiments** (Appendix C): Additional ablations and analyses as well as learning curves for single-stage tasks and Meta-World.
- **PSL Implementation Details** (Appendix D): Full details on how PSL is implemented, specifically the Sequencing Module.
- **Baseline Implementation Details** (Appendix E): Full details regarding baseline implements (E2E, RAPS, MoPA-RL, TAMP, SayCan)
- **Tasks** (Appendix F): Visualizations of each task as well as descriptions of each environment suite.
- **LLM Prompts and Plans** (Appendix G): Prompts that we use for our method as well as generated plans by the LLM.
- **Related Work** (Appendix H): Complete description of the related work.

## B  Ethics, Impacts and Limitations

### B.1  Ethical Considerations

There exist potential ethical concerns from the use of large-scale language models trained on internet-scale data. These models have been trained on vast corpi that may contain harmful content and implicit or even explicit biases expressed by internet users and may be capable of generating such content when queried. However, these issues are not specific to our work, rather they are inherent to LLMs trained at scale and other works that use LLMs face a similar ethical concern. Furthermore, we note that our research only makes use of LLMs to guide the behavior of a robot at a coarse level - specifying where a robot should go and how to leave the area. Our LLM prompting scheme ensures that this is all that is outputted from the LLM. Such outputs leave little scope for abuse, the LLM is not capable of performing the low-level control itself, which is learned through a task reward independently.

### B.2  Broader Impacts

Our research on guiding RL agents to solve long-horizon tasks using LLMs has potential for both positive and negative impacts. PSL draws connections between work on language modeling, motion planning and reinforcement learning for low-level control, which could lead to advancements in learning for robotics. PSL reduces the engineering burden on the human, instead of manually specifying/pre-training a library of behaviors, only a reward function and task description need be specified. More broadly, enabling robots to autonomously solve challenging robotics tasks increase the likelihood of robots one day being able to complete labor intensive work in dangerous situations. However, with increased automation, there are risks of potential job loss. Furthermore, with increased robot capabilities, there is a risk of misuse by bad actors, for which appropriate safeguards should be designed.

### B.3  Limitations

There are several limitations of PSL which leave scope for future work. 1) We impose a specific structure on the language plans and task solution (go to location X, interact there, so on). While this assumption covers a broad set of tasks as well illustrate in our experimental evaluation, tasks that involve interacting with multiple objects simultaneously or continuous switching between interaction and movement in a fluid manner may not be directly applicable. Future work can explore integrating a more expressive plan structure with the Sequencing Module. 2) Use of motion-planning makes application to dynamic tasks challenging. To that end, research on motion-planner distillation, such as Motion Policy Networks [47] could enable much faster, reactive behavior. 3) Although the RL agent is capable of adapting pose estimation errors, in the current formulation, there is not much the Learning Module can do if the high-level plan itself is entirely incorrect, or if the Sequencing module misinterprets the language instruction and moves the robot to the wrong object. One extension to address this limitation would be to fine-tune the Plan and Seq Modules online using RL as well, to adapt the large models to the specific environment and reward function.

## C   Additional Experiments

We perform additional analyses of PSL in this section.

|        | $\sigma = 0$ | $\sigma = 0.01$ | $\sigma = 0.025$ | $\sigma = 0.1$ | $\sigma = 0.5$ |
|--------|--------------|-----------------|------------------|----------------|----------------|
| **SayCan** | $1.0 \pm 0.0$ | $.93 \pm .05$ | $.27 \pm .12$ | $0.0 \pm 0.0$ | $0.0 \pm 0.0$ |
| **PSL**    | $1.0 \pm 0.0$ | $\mathbf{1.0 \pm 0.0}$ | $\mathbf{1.0 \pm 0.0}$ | $\mathbf{.75 \pm .07}$ | $0.0 \pm 0.0$ |

Table C.1: **Noisy Pose Ablation Results.** We add noise sampled from $\mathcal{N}(0, \sigma)$ to the pose estimates and evaluate SayCan and PSL. PSL is able to handle noisy poses by training online with RL, only observing performance degradation beyond $\sigma = 0.1$.

**PSL leverages stage termination conditions to learn faster.** While the target object sequence is necessary for PSL to plan to the right location at the right time, in this experiment we evaluate if knowledge of the stage termination conditions is necessary. Specifically, on the RS-Can task, we evaluate the use of stage termination condition checks in PSL to trigger the next step in the plan versus simply using a timeout of 25 steps. We find that it is in fact critical to use stage termination condition checks to enable the agent to effectively sequence the plan; use of a timeout results in dithering behavior which slows down learning. After 10K episodes we observe a performance improvement of 31% (100% vs. 69%) by including plan stage termination conditions in our pipeline.

**PSL produces policies that are robust to noisy pose estimates.** In real world settings, there is often noise in pose estimation due to noisy depth values, imperfect camera calibration or even network prediction errors. Ideally, the agent should be adapt to such potential failure modes: open-loop planning methods such as TAMP and SayCan are not well-suited to do so because they do not improve online. In this experiment we evaluate the PSL's ability to handle noisy/inaccurate poses by leveraging online interaction via RL. On the RS-Can task, we add zero-mean Gaussian noise to the pose, with $\sigma \in 0.01, 0.025, .1, .5$ and report our results in Table. C.1. While SayCan struggles to handle $\sigma > 0.01$, PSL is able to learn with noisy poses at $\sigma = .1$, at the cost of slower learning performance. Neither method performs well at $\sigma = 0.5$, however at that point the poses are not near the object and the effect is similar to resetting to a random robot pose in the workspace every episode.

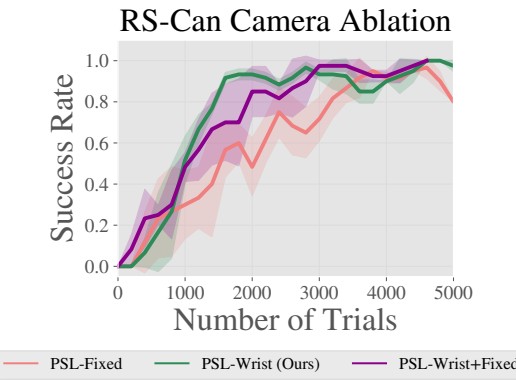

Figure C.1: **Camera View Learning Performance Ablation.** wrist camera views clearly accelerate learning performance, converging to near 100% performance **4x** faster than using fixed-view and **3x** faster than using wrist+fixed-view observations.

**Effect of camera view on policy learning performance:** As discussed in Sec. 2, for PSL we use local observations to improve learning performance and generalization to new poses. We validate this claim on the Robosuite Can task, in which we hypothesize that the local wrist camera view will accelerate policy learning performance. This is because the image of the can will be independent of the can's position in general since the Sequencing Module will initialize the RL agent as close to the can as possible. As observed in Fig. C.1, this is indeed the case - PSL learns **4x** faster than using a fixed view camera in terms of the number of trials. We additionally test if combining wrist and fixed view inputs (a common paradigm in robot learning) can alleviate the issue, however PSL with wrist cam is still **3x** faster at solving the task.

**Effect of camera view on chaining pre-trained policies:** In this ablation, we illustrate another important effect of using local views, such as wrist cameras: ease of chaining pre-trained policies. Since we leverage motion planning to sequence between policy executions, chaining pre-trained policies is relatively straightforward: simply execute the Sequencing Module to reach the first region of interest, execute the first pre-trained policy till its stage termination condition is triggered, then call the Sequencing Module on the next region, and so on. However, to do so, it is also crucial that the observations do not change significantly, so that the inputs to the pre-trained policies are not out of distribution (OOD). If we use a fixed, global view of the scene, the overall scene will change as multiple policies are executed, resulting in future policy executions failing due to OOD inputs. In Table C.2, we observe this exact phenomenon, in which any version of PSL that is provided a fixed-view input fails to chain pre-trained policies effectively, while PSL with local (wrist) views only is able to chain pre-trained policies on every task, up to 5 stages.

|                    | K-Single-Task | K-MS-3      | K-MS-4      | K-MS-5      |
|--------------------|---------------|-------------|-------------|-------------|
| **PSL**-Wrist       | $1.0 \pm 0.0$ | $1.0 \pm 0.0$ | $1.0 \pm 0.0$ | $1.0 \pm 0.0$ |
| **PSL**-Fixed       | $1.0 \pm 0.0$ | $0.0 \pm 0.0$ | $0.0 \pm 0.0$ | $0.0 \pm 0.0$ |
| **PSL**-Wrist+Fixed | $1.0 \pm 0.0$ | $0.0 \pm 0.0$ | $0.0 \pm 0.0$ | $0.0 \pm 0.0$ |

Table C.2: **Chaining Pre-trained Policies Ablation.** PSL can leverage local views (wrist cameras) to chain together multiple pre-trained policies via motion-planning using the Sequencing Module. While PSL with each camera input is able to produce a capable single-task policy, chaining only works with wrist camera observations as the observations are kept in-distribution.

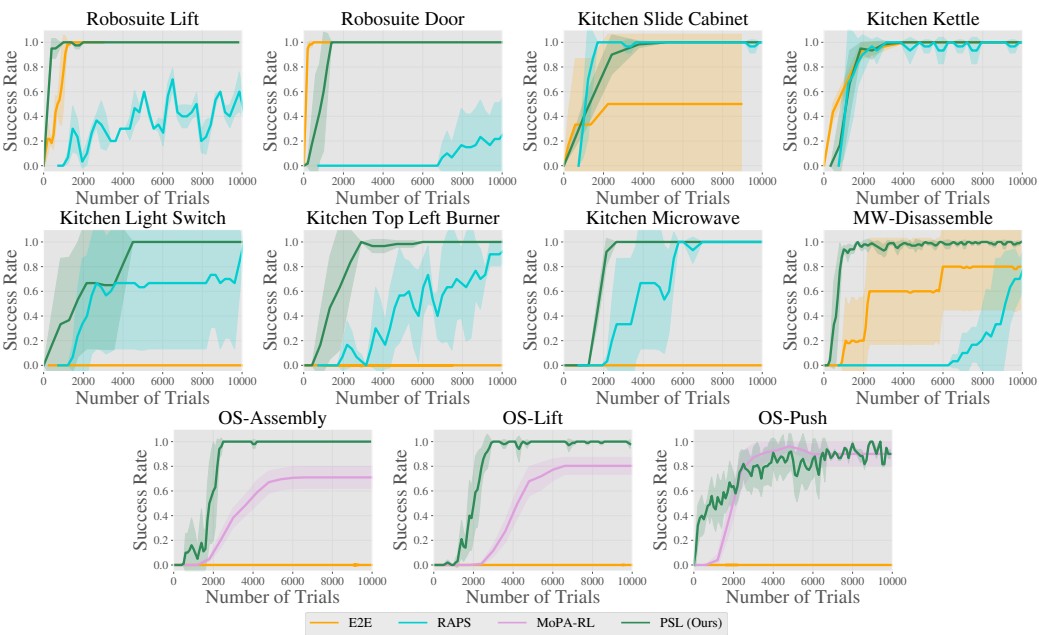

Figure C.2: **Single Stage Results.** We plot task success rate as a function of the number of trials. PSL improves on the efficiency of the baselines across single-stage tasks (*plan length of 1*) in Robosuite, Kitchen, Meta-World, and Obstructed Suite, **achieving an asymptotic success rate of 100% on all 11 tasks**.

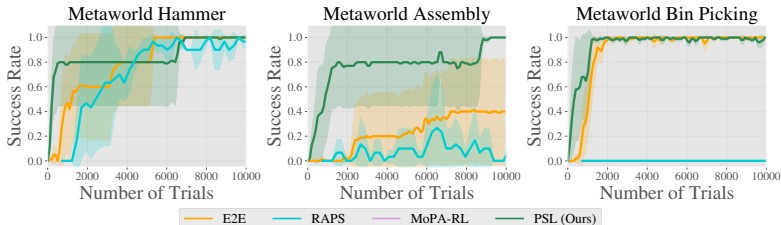

Figure C.3: **Meta-World Two Stage Learning Curves.** We plot task success rate as a function of the number of trials. PSL learns faster than the baselines by employing high-level planning to accelerate RL performance.

|        | MW-BinPick    | MW-Assembly   | MW-Hammer     |
|--------|---------------|---------------|---------------|
| **E2E**    | $1.0 \pm 0.0$ | $0.4 \pm 0.5$ | $0.0 \pm 1.0$ |
| **RAPS**   | $0.0 \pm 0.0$ | $0.3 \pm .25$ | $1.0 \pm 0.0$ |
| **TAMP**   | $1.0 \pm 0.0$ | $1.0 \pm 0.0$ | $0.0 \pm 0.0$ |
| **SayCan** | $1.0 \pm 0.0$ | $0.5 \pm .08$ | $1.0 \pm 0.0$ |
| **PSL**    | $1.0 \pm 0.0$ | $1.0 \pm 0.0$ | $1.0 \pm 0.0$ |

Table C.3: **Metaworld Two Stage Results.** While the baselines perform well on most of the tasks, only PSL is able to consistently solve every task. This is because the LLM planning and Sequencing modules ease the learning burden for the RL policy, enabling it to learn contact-rich, long-horizon behaviors.

## D    PSL Implementation Details

---

**Algorithm 1** Plan-Seq-Learn Overview

---

**Require:** LLM, Pose Estimator P, task description $g_l$, Motion Planner MP, low-level horizon $H_l$

    *Planning Module*

    High-level plan $\mathcal{P} \leftarrow$ Prompt(LLM, $g_l$)

    **for** $p \in \mathcal{P}$ **do**

    *Sequencing Module*

        target region ($t$), termination condition $\leftarrow p$

        Compute pose $q_{target} = P(O_t^{global}, t)$

        Achieve pose MP($q_{target}, O_t^{global}$)

    *Learning Module*

        **for** $i = 1, ..., H_l$ **do**

            Get action $a_t \sim \pi_\theta(O_t^{local})$

            Get next state $O_{t+1}^{local} \sim p(|s_t, a_t)$.

            Store $(O_t^{local}, a_t, O_{t+1}^{local}, r)$ into $\mathcal{R}$

            update $\pi_\theta$ using RL

            **if** stage termination condition **then**

                break

            **end if**

        **end for**

    **end for**

---

### D.1    Planning Module

Given a task description $g_l$, we prompt an LLM using the format described in Sec. 2.4 to produce a language plan. We experimented with a variety of publicly available and closed-source LLMs including LLAMA [48], LLAMA-2 [49], GPT-3 [50], Chat-GPT, and GPT-4 [34]. In initial experiments, we found that GPT-based models performed best, and GPT-4 in particularly most closely adhered to the prompt and produced the most accurate plans. As a result, in our experiments, we use GPT-4 as the LLM planner for all tasks. We sample from the model with temperature 0 for determinism. Sometimes, the LLM hallucinates non-existent stage termination conditions or objects. As a result, we add a pre-processing step in which we delete components of the plan that contain such hallucinations.

### D.2    Sequencing Module

The input to the Sequencing Module is $O^{global}$. In our experiments, we use two camera views, $O_1^{global}$ and $O_2^{global}$, which are RGB-D calibrated camera views of the scene, to obtain unoccluded views of the scene. We additionally provide the current robot configuration, which is joint angles for robot arms: $q_{joint}$ and the target region label around which the RL policy must perform environment interaction. From this information, the module must solve for a collision free path to a region near the target. This problem can be addressed by classical motion planning. We take advantage of sampling-based motion planning due to its minimal setup requirements (only collision-checking) and favorable performance on planning. In order to run the motion planner, we require a collision checker, which we implement using point-clouds. To compute the target object position, we use predicted segmentation along with calibrated depth, as opposed to a dedicated pose estimation network, primarily because state of the art segmentation models [27, 28] have significant zero-shot capabilities across objects.

**Projection:** In this step, we project the depth map from each global view of the scene, $O_1^{global}$ and $O_2^{global}$ into a point-cloud $PC^{global}$ using their associated camera matrices $K_1^{global}$ and $K_2^{global}$. We perform the following processing steps to clean up $PC^{global}$: 1) cropping to remove all points outside the workspace 2) voxel down-sampling with a size of 0.005 $m^3$ to reduce the overall size of $PC^{global}$ 3) outlier removal, which prunes points that are farther from their 20 neighboring points than the average in the point-cloud as shown in Fig. D.1.

**Algorithm 2** PSL Implementation

**Require:** LLM, task description $g_l$, Motion Planner MP, low-level horizon $H_l$, segmentation model $\mathcal{S}$, RGB-D global cameras, RGB wrist camera, Camera Matrix $K^{global}$
1: initialize RL: $\pi_\theta$, replay buffer $\mathcal{R}$
    *Planning Module*
2: High-level plan $\mathcal{P} \leftarrow$ Prompt(LLM, $g_l$)
3: **for** episode $1...N$ **do**
4:     **for** $p \in \mathcal{P}$ **do**
    *Sequencing Module*
5:         target region $(t)$, termination condition $\leftarrow p$
6:         $PC^{global} = $ Projection($O_1^{global}, O_2^{global}, K^{global}$)
7:         $M_{robot}, M_{obj} = $ Segmentation($O_1^{global}, O_2^{global}$, robot, object)
8:         $PC^{robot}, PC^{object} = M_{robot} * PC^{global}, M_{obj} * PC^{scene}$
9:         $PC^{scene} = PC^{global} - PC^{robot}$
10:        $ee_{target} = $ mean($PC^{obj}$)
11:        $q_{target} = $ IK($ee_{target}$)
12:        MotionPlan(MP, $q_{target}, PC^{scene}$)
    *Learning Module*
13:        **for** $i = 1, ..., h$ low-level steps **do**
14:           Get action $a_t \sim \pi_\theta(O_t^{local})$
15:           Get next state $O_{t+1}^{local} \sim p(\cdot|s_t, a_t)$.
16:           Store $(O_t^{local}, a_t, O_{t+1}^{local}, r)$ into $\mathcal{R}$
17:           Sample $(O_k^{local}, a_t, O_{k+1}^{local}, r) \sim \mathcal{R}$                $\triangleright$ k = random index
18:           update $\pi_\theta$ using RL
19:           **if** post-condition **then**
20:              break
21:           **end if**
22:        **end for**
23:     **end for**
24: **end for**

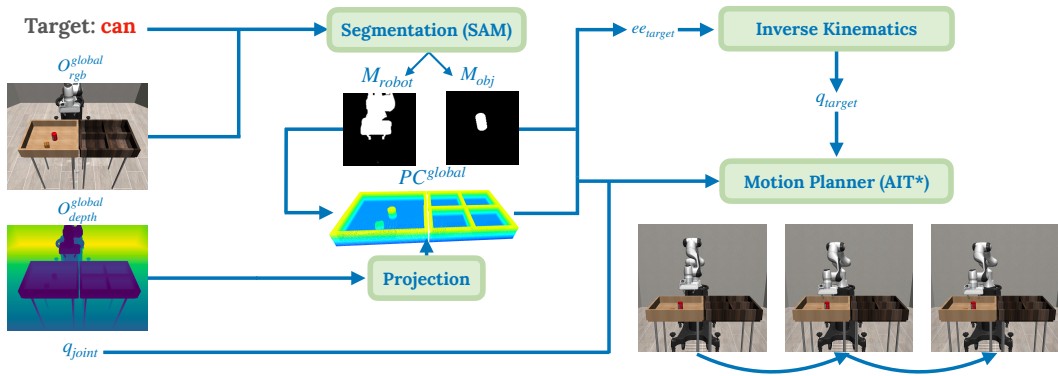

Figure D.1: **Sequencing Module.** Inputs to the Sequencing Module are two calibrated RGB-D fixed views, $O^{global}$, the proprioception $q_{joint}$ and the target object. It performs visual motion planning to the target object by computing a scene point-cloud ($PC^{global}$), segmenting the target object ($M_{obj}$) to estimate its pose ($q_{target}$), segmenting the robot ($M_{robot}$) to remove it from $PC^{global}$ and motion planning using AIT*.

**Segmentation:** We compute masks for the robot ($M_{robot}$) and the target object ($M_{obj}$) by using a segmentation model (SAM [27]) $\mathcal{S}$ which segments the scene based on RGB input. We reduce noise in the masks by filling holes, computing contiguous mask clusters and selecting the largest mask. We use $M_{robot}$ to remove the robot from $PC^{global}$, in order to perform collision checking of the robot against the scene. Additionally, we use $M_{obj}$ along with $PC^{global}$ to compute the object point-cloud $PC^{obj}$, which we average to obtain an estimate of object position, which is the target position for the motion planner. For the manipulation tasks we consider in the paper, this is the target end-effector pose of the robot, $ee_{target}$.

**Visual Motion Planning:** Given the target end-effector pose $ee_{target}$, we use inverse kinematics (IK) to compute $q_{target}$ and pass $q_{joint}, q_{target}, PC^{global}$ into a joint-space motion planner. To that end, we use a sampling-based motion planner, AIT* [38], to perform motion planning. In order to implement collision checking from vision, for a sampled joint-configuration $q_{sample}$, we compute the corresponding position of the robot mesh and compute the occupancy of each point in the scene point-cloud against the robot mesh. If the object is detected as grasped, then we additionally remove the object from the scene pointcloud, compute its convex hull and use the signed distance function of the joint robot-object mesh for collision checking. As a result, the Sequencing Module operates entirely over visual input, and achieves a pose near the region of interest before handing off control to the local RL policy. We emphasize that the Sequencing Module *does not need to be perfect*, it merely needs to produce a reasonable initialization for the Learning Module.

## D.3 Learning Module

### D.3.1 Stage Termination Details

As described in Section 2, we use stage termination conditions to determine when the Learning Module should hand control back to the Sequencing Module to continue to the next stage in the plan. For the tasks we consider, these stage termination conditions amount to checking for a grasp or placement for the target object in the stage. For example, for RS-NutRound, the plan for the first stage is (grasp, nut) and the plan for the second stage is (place, peg). Placements are straightforward to check: simply evaluate if the object being manipulated is within a small region near the target object. This can be computed using the estimated pose of the two objects (current and target). Grasps are more challenging to estimate and we employ a two stage pipeline to detecting a grasp. First, we estimate the object pose and then evaluate if the z value has increased from when the stage began. Second, in order to ensure the object is not simply tossed in the air, we check if the robot's gripper is tightly caging the object. We do so by collision checking the object point-cloud against the gripper mesh. We use the same collision checking procedure as outlined in Sec 2 for checking collision between the scene point-cloud and robot mesh.

### D.3.2 Training Details

For all tasks, we use the reward function defined by the environment, which may be dense or sparse depending on the task. We find that for PSL, it is crucial to use an action-repeat of 1, in general we found that increasing this harmed performance, in contrast to the E2E baseline which performs best with an action repeat of 2. For training policies using DRQ-v2, we use the default hyper-parameters from the paper, held constant across all tasks. We train policies using 84x84 images. We use the "medium" difficult exploration schedule defined in [39], which anneals the exploration $\sigma$ from 1.0 to 0.1 over the course of 500K environment steps. Due to memory concerns, instead of using a replay buffer size of 1M as done in Yarats et al. [39], ours is of size 750K across each task. Finally, for path length, we use the standard benchmark path length for E2E and MoPA-RL, 5 per stage for RAPS following Dalal et al. [45], and 25 per stage for PSL.

# E  Baseline Implementation Details

## E.1  RAPS

For this baseline, we simply take the results from the RAPS [45] paper as is, which use Dreamer [51] and sparse rewards. In initial experiments, we attempted to combine RAPS with DRQ-v2 [39] and found that Dreamer performed better, which is consistent with RAPS+Dreamer having the best results in Dalal et al. [45]. We additionally tried to run RAPS with dense rewards, but found that the method performed significantly worse. One potential reason for this is that it is not clear exactly how to aggregate the dense rewards across primitive executions - we tried simply taking the dense reward after executing a primitive as well as simply summing the rewards of intermediate primitive executions. Both performed worse than training RAPS with sparse rewards. Note that PSL outperforms RAPS even when both methods have only access to sparse rewards, e.g. the Kitchen environments. We observe clear benefits over RAPS on the single-stage (Fig. C.2) and multi-stage (Table 2) tasks.

## E.2  MoPA-RL

As described in the main paper, we take the results from MoPA-RL [41] as is on the Obstructed Suite of tasks. Those results were run from state-based input and leveraged the simulator for collision checking. We do so as we were unable to successfully combine MoPA-RL with DRQ-v2 based on the publicly released implementations of both methods.

## E.3  TAMP

We use PDDLStream [46] as the TAMP algorithm of choice as it has been shown to have strong planning performance on long-horizon manipulation tasks in Robosuite [52, 53]. The PDDLStream planning framework models the TAMP domain and uses the adaptive algorithm, a sampling based algorithm, to plan. This TAMP method uses samplers for grasp generation, placement sampling, inverse kinematics, and motion planning, making performance stochastic. Hence we average performance across 50 evaluations to reduce variance. We adapt the authors TAMP implementation (from [52, 53]) for our tasks. Note this method uses privileged access to the simulator, leveraging knowledge about the task (which must be explicitly specified in a problem file), the scene (from the domain file and access to collision checking) and 3D geometry of the environment objects.

## E.4  SayCan

As described in the main paper, we re-implement SayCan Ahn et al. [1] using GPT-4 (the same LLM we use in our methdo) and manually engineered pick/place skills that use pose-estimation and motion-planning. Following our Sequencing module: 1) we build a 3D scene point-cloud using camera calibration and depth images 2) we perform vision-based pose estimation using segmentation along with the scene point cloud and 3) we run motion planning using collision queries from the 3D point-cloud, which is used for collision queries. Finally, we use heuristically engineered pick and place primitives to perform interaction behavior which we describe as follows. We note that for our tasks of interest, the pick motion can be represented as a top-grasp. Once we position the robot near the object; we then simply lower the robot arm till the end-effector (not the grippers) come in contact with the object. We then close the gripper to execute the grasp. For place, we follow the implementation of Ahn et al. [1] and lower the held object until contact with a surface, then release (open the gripper) and lift the robot arm. We set the affordance function for both skills to 1, following the design in Ahn et al. [1] for motion planned skills.

For LLM planning, we specify the following prompt:

> Given the following library of robot skills: ... Task description: ... Make sure to take into account object geometry. Formatting of output: a list of robot skills. Don't output anything else.

This prompt is the same as our prompt except we specify the robot skill library in terms of object centric behaviors, instead of stage termination conditions.

> Given the following library of robot skills: ... Task description: ... Give me a simple plan to solve the task using only the provided skill library. Make sure the plan follows the formatting specified below and make sure to take into account object geometry. Formatting of output: a list of robot skills. Don't output anything else.

Robosuite

> **Skill Library:** pick can, pick milk, pick cereal, pick bread slice, pick silver nut, pick gold nut, put can on/in X, put milk on/in X, put cereal on/in X, put bread slide on/in X, put silver nut on/in X, put gold nut on/in X, grasp door handle, turn door handle, pick cube

Kitchen

> **Skill Library:** grasp vertical door handle for slide cabinet, move left, move right, grasp hinge cabinet, grasp top left burner with red tip, rotate top left burner with red tip 90 degree clockwise, rotate top left burner with red tip 90 degrees counterclockwise, push light switch knob left, push light switch knob right, grasp kettle, lift kettle, place kettle on/in X, grasp microwave handle, pull microwave handle

Metaworld:

> **Skill Library:** grasp cube, place cube on/in X, grasp hammer, place hammer, hit nail with hammer, grasp wrench, lift wrench

Obstructed-Suite

> **Skill Library:** grasp can, place can in bin, insert table leg in X, move table leg, grasp cube, place cube on table, push cube

 # F  Tasks

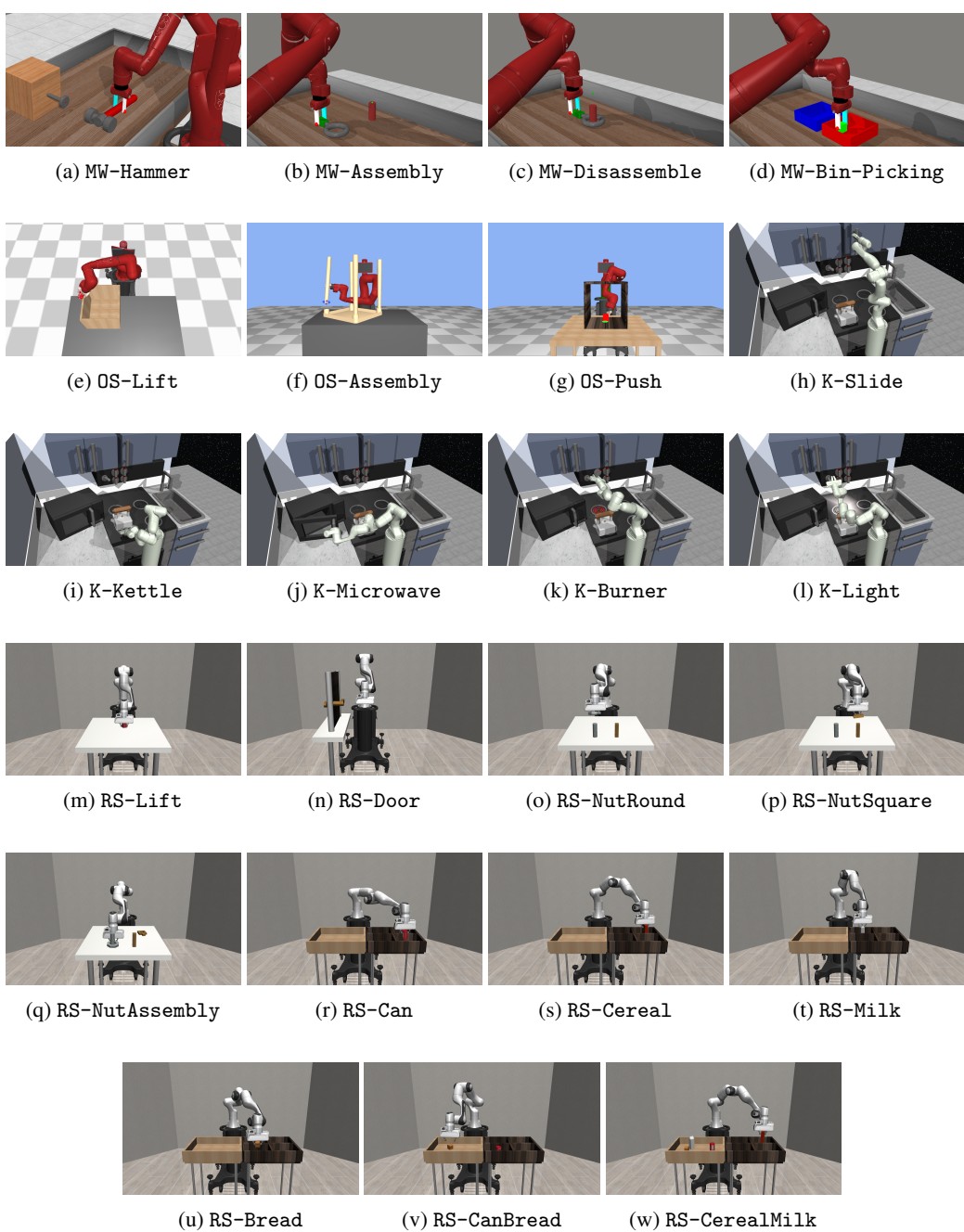

(a) MW-Hammer  (b) MW-Assembly  (c) MW-Disassemble  (d) MW-Bin-Picking

(e) OS-Lift  (f) OS-Assembly  (g) OS-Push  (h) K-Slide

(i) K-Kettle  (j) K-Microwave  (k) K-Burner  (l) K-Light

(m) RS-Lift  (n) RS-Door  (o) RS-NutRound  (p) RS-NutSquare

(q) RS-NutAssembly  (r) RS-Can  (s) RS-Cereal  (t) RS-Milk

(u) RS-Bread  (v) RS-CanBread  (w) RS-CerealMilk

Figure F.1: **Task Visualizations**. PSL is able to solve all tasks with at least 80% success rate from purely visual input.

We discuss each of the environment suites that we evaluate using PSL. All environments are simulated using the MuJoCo simulator [54].

1. **Meta-World** (Row 1 of Fig. F.1). Meta-World, introduced by Yu et al. [40], aims to offer a standardized suite for multi-task and meta-learning methods. The benchmark consists of 50 separate manipulation tasks with a Sawyer robot, well-shaped reward functions, involve manipulating a single object to a randomized goal position, or multiple objects to a deterministic goal position. We evaluate on the single-task, multi-goal, v2 variants of the Meta-World environments. All environments use end-effector position control - a 3DOF arm action space along with gripper control - orientation is fixed. In our evaluation we use the default environment task rewards, a fixed camera view for the baselines and a wrist camera for our local policies. We refer the reader to the Meta-World paper for additional details regarding the environment suite.

2. **Obstructed Suite** (Rows 1-2 of Fig. F.1). The Obstructed Suite of tasks introduced by Ya-mada et al. [41] are a challenging set of tasks requiring a Sawyer arm to perform obstacle avoidance while solving the task. The `OS-Lift` task requires the agent to pick up a can that is inside a tall box, requiring it to reach over the walls to grab the object and then lift it without making contact with the edges of the bin. The `OS-Push` environment tasks the agent with push a block to the goal in the present of a bin that forces the agent to adjust its motion in order to avoid being blocked by its upper joints. Finally, the `OS-Assembly` task involves moving the robot arm to a precise placement location while avoiding obstacles, then performing the table leg placement. Note that we evaluate our method on these environments from visual input, a more challenging setting than the one considered by Yamada et al. [41].

3. **Kitchen** (Rows 2-3 of Fig. F.1). The Kitchen manipulation suite introduced in the Relay Policy Learning paper [42] and maintained in D4RL [43] is a set of challenging, sparse reward, joint-controlled manipulation tasks in a single kitchen. The tasks require the ability to explore efficiently whilst also being able to chain skills across long temporal horizons, to achieve behaviors such as opening the microwave, moving the kettle, flicking the light switch, turning the burner, and finally sliding the cabinet door (`K-MS-5`). Aside from the single-stage tasks described in Section 3, we evaluate on three multi-stage tasks which require chaining the single-stage tasks in a particular order. `K-MS-3` involves moving the kettle, flicking the light switch and turning the burner, while `K-MS-4` is the same as `K-MS-3`, but the agent must first open the microwave door then execute the rest of the tasks.

4. **Robosuite** (Rows 3-6 of Fig. F.1). The Robosuite benchmark from Zhu et al. [44] contains challenging, long-horizon manipulation tasks involving pick-place and nut assembly, as well as simpler tasks that involve lifting a cube and opening a door. The rewards are coarsely defined in terms of distances to targets as well as grasp/placement conditions, which, in fact, are straightforward to implement in the real world as well using pose estimation. This stands in contrast to Meta-World which spends considerable engineering effort defining well-shaped dense rewards often by taking advantage of object geometry. As a result, learning-based methods struggle to make any progress on Robosuite tasks that involve more than a single-stage - optimizing the reward function tends to leave the agent a local minima. The suite also contains a well-tuned, realistic Operation Space Control [55] implementation that we leverage to train policies in end-effector space.

 # G   LLM Prompts and Plans

In this section, we list the LLM prompts per task.

Overall prompt structure:

> Stage termination conditions: (grasp, place). Task description: ... Give me a simple plan to solve the task using only the stage termination conditions. Make sure the plan follows the formatting specified below and make sure to take into account object geometry. Formatting of output: a list in which each element looks like: (<object/region>, <operator>). Don't output anything else.

## G.1   Robosuite

`RS-PickPlaceCan`:

> **Task Description** can goes into bin 1.
> **Plan:** [("can", "grasp"), ("bin 1", "place")])

`RS-PickPlaceCereal`:

> **Task Description:** cereal goes into bin 3.
> **Plan:** [("cereal", "grasp"), ("bin 3", "place")])

`RS-PickPlaceMilk`:

> **Task Description**: milk goes into bin 2.
> **Plan:** [("milk", "grasp"), ("bin 2", "place")])

`RS-PickPlaceBread`:

> **Task Description:** bread slice goes into bin 4.
> **Plan:** [("bread slice", "grasp"), ("bin 4", "place")])

`RS-PickPlaceCanBread`:

> **Task Description:** can goes into bin 1, bread slice in bin 4.
> **Plan:** [("can", "grasp"), ("bin 1", "place"), ("bread slice", "grasp"), ("bin 4", "place")])

`RS-PickPlaceCerealMilk`:

> **Task Description:** milk goes into in bin 2, cereal in bin 3.
> **Plan:** [("cereal", "grasp"), ("bin 3", "place"), ("milk", "grasp"), ("bin 2", "place")])

`RS-NutAssembly`:

> **Task Description:** The silver nut goes on the silver peg and the gold nut goes on the gold peg.
> **Plan:** [("silver nut", "grasp"), ("silver peg", "place"),("gold nut", "grasp"), ("gold peg", "place")]

790 `RS-NutAssemblySquare:`

> **Task Description:** The gold nut goes on the gold peg.
> **Plan:** [("gold nut", "grasp"), ("gold peg", "place")]

791 `RS-NutAssemblyRound:`

> **Task Description:** The silver nut goes on the silver peg.
> **Plan:** [("silver nut", "grasp"), ("silver peg", "place")]

792 `RS-Lift:`

> **Task Description:** lift the red cube.
> **Plan:** [("red cube", "grasp")]

793 `RS-Door:`

> **Task Description:** open the door.
> **Plan:** [("door handle", "grasp")]

794 ## G.2 Meta-World

795 `MW-Assembly:`

> **Task Description:** put the green wrench on the maroon peg.
> **Plan:** [("green wrench", "grasp"), ("maroon peg", "place")]

796 `MW-Disassemble:`

> **Task Description:** remove the green wrench from the peg.
> **Plan:** [("green wrench", "grasp")]

797 `MW-Hammer:`

> **Task Description:** use the red hammer to push in the nail.
> **Plan:** [("red hammer", "grasp"), ("nail", "push")]

798 `MW-Bin-Picking:`

> **Task Description:** move the cube in the red bin into the blue bin.
> **Plan:** [("cube in red bin", "grasp"), ("blue bin", "place")]

799 ## G.3 Kitchen

800 `Kitchen-Microwave:`

> **Task Description:** open the microwave door.
> **Plan:** [("microwave door handle", "grasp")]

801 `Kitchen-Slide`

> **Task Description:** use the rightmost vertical bar to slide open the door.
> **Plan:** [("rightmost vertical bar", "grasp")]

802 `Kitchen-Light`

> **Task Description:** use the round knob to turn on the light.
> **Plan:** [("knob", "grasp")]

803 `Kitchen-Burner`

> **Task Description:** turn the top left burner with the red tip.
> **Plan:** [("top left burner with the red tip", "grasp")]

804 `Kitchen-Kettle`

> **Task Description:** move the kettle forward.
> **Plan:** [("kettle", "grasp")]

805 **G.4 Obstructed Suite**

806 `OS-Lift:`

> **Task Description:** lift red can from wooden bin.
> **Plan:** [("red can', "grasp")]

807 `OS-Assembly:`

> **Task Description:** move the table leg, which is already in your hand, into the empty hole.
> **Plan:** [("empty hole', "place")]

808 `OS-Push:`

> **Task Description:** push the red block onto the green circle.
> **Plan:** [("red block", "grasp")]

# H  Related Work

**Classical Approaches to Long Horizon Robotics:** Historically, robotics tasks have been approached via the Sense-Plan-Act (SPA) pipeline [56, 57, 58, 59, 60], which requires comprehensive understanding of the environment (sense), a model of the world (plan), and a low-level controller (act). Traditional approaches range from manipulation planning [61, 62], grasp analysis [63], and Task and Motion Planning (TAMP) [64], to modern variants incorporating learned vision [65, 66, 67]. Planning algorithms enable long horizon decision making over complex and high-dimensional action spaces. However, these approaches can struggle with contact-rich interactions [68, 69], experience cascading errors due to imperfect state estimation [70], and require significant manual engineering and systems effort to setup [71]. Our method leverages learning at each component of the pipeline to sidestep these issues: it handles contact-rich interactions using RL, avoids cascading failures by learning online, and sidesteps manual engineering effort by leveraging pre-trained models for vision and language.

**Planning and Reinforcement Learning:** Recent work has explored the integration of motion planning and RL to combine the advantages of both paradigms [72, 41, 73, 74, 75, 76, 77]. GUAPO Lee et al. [72] is similar to the Seq-Learn components of our method, yet their system considers the single-stage regime and is focused on keeping the RL agent in areas of low pose-estimator uncertainty. Our method instead considers long-horizon tasks by encouraging the RL agent to follow a high-level plan given by an LLM using vision-based motion planning. MoPA-RL [41] also bears resemblance to our method, yet it opts to learn when to use the motion planner via RL, requiring the RL agent to discover the right decomposition of planner vs. control actions on its own. Furthermore, roll-outs of trajectories using MoPA can result in the RL agent choosing to motion plan multiple times in sequence, which is inefficient - one motion planner action is sufficient to reach any position in space. In our method, we instead explicitly decompose tasks into sequences of contact-free reaching (motion planner) and contact-rich environment interaction (RL).

**Language Models for RL and Robotics** LLMs have been applied to RL and robotics in a wide variety of ways, from planning [1, 2, 14, 3, 4, 17, 18, 19], reward definition [20, 21], generating quadrupedal contact-points [22], producing tasks for policy learning [23, 24] and controlling simulation-based trajectory generators to produce diverse tasks [25]. Our work instead focuses on the online learning setting and aims to leverage language model driven planning to guide RL agents to solve new robotics tasks in a sample efficient manner. BOSS Zhang et al. [26] is closest to our overall method; this concurrent work also leverages LLM guidance to learn new skills via RL. Crucially, their method depends on the existence of a skill library and learns skills that are combination of high-level actions. Our method instead efficiently learns *low-level* robot control skills without depending on a pre-defined skill library, by taking advantage of motion planning to track an LLM plan.

