# OpenReview forum: "Plan-Seq-Learn: Language Model Guided RL for Solving Long Horizon Robotics Tasks"
_robot-learning.org/CoRL/2023/Workshop/TGR — CoRL 2023 Workshop TGR Poster_

### Official Review · Reviewer_16vG · 2023-10-19
**Accept**

**Rating:** 7
**Confidence:** 4

**Review:**

The motivation of using RL to strengthen skill learning for LLM-based manipulation is clear. However, there might be missing key components in the connection between LLM and RL-training. Maybe LLM for reward generation is a potential way? Hope to see more discussions over here. Overall great work!

---

### Decision · Program_Chairs · 2023-10-20

**Decision:**

Accept (Poster)

**Comment:**

Great paper and closely aligned topic!